# Allogeneic Hematopoietic Stem Cell Transplantation Despite Severe *Fusarium solani* Infection in a Lung Transplanted Patient—A Case Report

**DOI:** 10.3390/microorganisms13040703

**Published:** 2025-03-21

**Authors:** Monica Tozzi, Adele Santoni, Marta Franceschini, Margherita Malchiodi, Irene Bernareggi, Beatrice Esposito Vangone, Corrado Zuanelli Brambilla, Elisabetta Zappone, Mariapia Lenoci, Francesca Toraldo, Valeria Del Re, Alice Pietrini, Elena Marchini, Antonella Fossi, David Bennett, Elena Bargagli, Giuseppe Marotta, Alessandro Bucalossi, Monica Bocchia

**Affiliations:** 1Cellular Therapy and Blood Establishment Unit, Azienda Ospedaliero-Universitaria Senese, 53100 Siena, Italy; m.tozzi@ao-siena.toscana.it (M.T.);; 2Hematology Unit, Azienda Ospedaliero-Universitaria Senese, University of Siena, 53100 Siena, Italy, marghe.mal95@gmail.com (M.M.); dottoressabernareggi@gmail.com (I.B.); b.espositovangone@student.unisi.it (B.E.V.); corrado.zuanelli@ao-siena.toscana.it (C.Z.B.); ezappone@gmail.com (E.Z.); bocchia@unisi.it (M.B.); 3Apheresis Unit, Blood Transfusion Service, Azienda Ospedaliero-Universitaria Senese, 53100 Siena, Italy; 4Respiratory Diseases, Azienda Ospedaliero-Universitaria Senese, University of Siena, 53100 Siena, Italy; antonella.fossi@gmail.com (A.F.); bargagli2@unisi.it (E.B.)

**Keywords:** *Fusarium solani*, fungal infection, allogeneic hematopoietic stem cell transplantation, lung transplant, acute myeloid leukemia

## Abstract

Solid-organ transplant patients require prolonged immunosuppression, increasing their risk of hematologic disorders. For these conditions, allogeneic hematopoietic stem cell transplantation (HSCT) is a potential treatment, but it carries significant risk of treatment-related mortality due to the high possibility of developing rare infectious complications. We report a case of a 55-years-old male with a history of bilateral lung transplantation for extrinsic allergic alveolitis in 2015, who developed acute myeloid leukemia/myelodysplastic syndrome (AML/MDS) with TP53 mutation seven years later. During induction therapy, he experienced systemic fungal infection caused by *Fusarium solani* and he underwent HSCT conditioning with active intravitreal fungal infection. It is noteworthy that cases of patients undergoing HSCT after a prior lung transplant are exceedingly rare. The medical literature primarily documents cases where HSCT is performed first, followed by lung complications. Cases with the opposite timeline are extremely uncommon, and there is limited data on their outcomes; thus, the patient depicted here may help management and decision making of physicians facing this rare sequence of diseases and treatments.

## 1. Introduction

Infectious complications are one of the most significant risks in the management of hematological patients. Despite considerable advancements, these complications still represent a critical challenge due to patients’ fragility. Over time, hematologists have implemented well-defined protocols for early and effective management of febrile neutropenia, reducing infectious related mortality (IRM) [1].

However, the clinical infectious landscape frequently presents complicated scenarios that defy standard therapeutic approaches. These challenging cases often need a more nuanced decision-making process, as conventional treatments may not be effective.

Moreover, in specific scenarios, our ability to manage these difficulties is further complicated by the nature of the immune system itself. While the immune system is typically our greatest ally in fighting infections, sometimes it can paradoxically become a source of additional challenges [2].

We present the case of a 55-year-old male who received an allogeneic hematopoietic stem cell transplant (HSCT) from an HLA-matched sibling donor for acute myeloid leukemia/myelodysplastic syndrome (AML/MDS) with TP53 mutation, seven years after a lung transplant. This represents the second documented case of this dual transplant but the first to be complicated by a rare and aggressive fungal infection [3] occurring during pre-transplant induction treatment. The infection, caused by *Fusarium solani*, involved the skin and soft tissues, progressed to disseminated disease confirmed by positive blood cultures, and showed additional fungal localization—presumably of the same pathogen—at the ocular level. The patient remained under active treatment with intravitreal injections during transplantation. This condition required a careful balance between tailoring the immunosuppressive therapy for solid-organ rejection prevention, reducing the risk related to infection, and maintaining the graft-versus-leukemia effect in the context of a non-myeloablative conditioning regimen.

Invasive mold infections (IMIs) are a known risk for patients with hematological malignancies and other immunosuppressing conditions. Despite routine primary antifungal prophylaxis in immunocompromised individuals, including those with acute myeloid leukemia (AML) receiving chemotherapy, rare IMIs still occur. The incidence of *Fusarium* spp. infections in patients with acute leukemia in Europe is 0.06% [4] and was 1.2% among 750 allogeneic HSCT recipients in the United States [5]. *Fusarium* infections occur primarily through direct inoculation via contaminated materials in skin or mucosal lesions or by airborne transmission through inhalation of fungal spores. In immunocompetent hosts, they typically manifest as keratitis or onychomycosis, while in immunocompromised individuals, they can progress to severe disseminated infections through angioinvasion and tissue destruction.

In immunocompromised patients, *Fusarium* infections often manifest as persistent fever, sinusitis, pneumonia, or skin infections like cellulitis, with lung involvement occurring in about half of cases. In nearly 70% of these high-risk patients, the infection occurs in the most severe disseminated form, and it is characterized by a fever lasting over ten days during profound neutropenia, accompanied by skin lesions and blood cultures positive for yeast [6,7,8,9,10,11,12,13].

## 2. Case History

In April 2015, our patient received a bilateral lung transplant following severe extrinsic allergic alveolitis due to probable exposure to exogenous toxic substances at the age of 47. Immunosuppressive therapy was set with Tacrolimus, Mycophenolate Mofetil (MMF), and corticosteroids. During pneumological follow-up, *Streptococcus pneumoniae*, *Pseudomonas aeruginosa*, and increased galactomannan antigen were detected by bronchoalveolar lavage; thus, antifungal therapy with Isavuconazole was started for the severe patient’s immunosuppresion, despite the fact he was asymptomatic.

In April 2022, he contracted SARS-CoV-2. At the end of the infection, he displayed progressive pancytopenia (hemoglobin (Hb) decreased from 13 g/dL to 9.5 g/dL and white blood cells (WBC) from 4700/mmc to 2500/mmc). Suspecting drug toxicity, MMF was discontinued, and steroids were increased.

In July 2022, pancytopenia further worsened, and therefore, our patient was referred to hematologic counseling, presenting Hb 9 g/dL, MCV 112 fL, WBC 990/mmc (neutrophils 10% and blasts 2%), and platelets (PLTs) 64,000/mmc. Bone marrow (BM) biopsy evaluation was performed displaying severe hypoplasia with a markedly reduced, non-maturing myeloid series and evidence of 15–19% myeloid blasts. FISH and karyotype did not show any alterations, and molecular analysis, performed by RT-PCR, revealed NPM1 and FLT3 wild-type and WT-1 3109 copies. Next-generation-sequencing molecular analysis found TP53 gene mutation (VAF 40.35%).

The patient was diagnosed with AML/MDS with a TP53 mutation.

The case was discussed in a multidisciplinary meeting with hematologists, HSCT specialists, and pulmonologists. Considering the good performance status and the high-risk hematological disease, induction therapy followed by allogeneic HSCT in the event of complete response (CR) was collectively indicated as a feasible therapeutic approach.

In August 2022, the patient was admitted to the Hematology Unit of Azienda Ospedaliera Universitaria Senese, starting chemotherapy (cht) with CPX-351 (intravenous (IV) cytarabine 100 mg/m^2^ and daunorubicin 44 mg/m^2^ on days 1, 3, and 5), showing an auto-limiting skin rash. Ongoing prophylaxis with Isavuconazole, due to previous galactomannan positivity, Azithromycin, Sulfamethoxazole/Trimethoprim, and Aciclovir was maintained. Immunosuppressive therapy with Tacrolimus was tapered until discontinuation, and Prednisone was continued at 5 mg from day 4 to 16.

Rectal swab on admission was positive for extended spectrum beta lactamase (ESBL)-producing *Escherichia coli*. The patient presented with a fever on day 5, starting broad-spectrum antibiotic therapy tailored to the results of the rectal swab cultures’ antibiogram, with Piperacillin/Tazobactam and Amikacin. Despite targeted antibiotics, negative blood cultures, and negative chest X-ray, the fever persisted, and antibiotic therapy was changed, introducing Vancomycin and Meropenem. Furthermore, a chest CT scan, echocardiography, abdominal ultrasound, and multiple microbiological and serological tests, including serum galactomannan, were performed without identification of infection.

On day 16, suspecting a non-infectious etiology of the fever, corticosteroid therapy was implemented with methylprednisolone 20 mg IV. Unfortunately, on day 18, hyphae and yeast-like cells identified as *Fusarium solani* were isolated in a new set of blood cultures. Since antimycogram showed azole resistance (voriconazole Minimum Inhibitory Concentration (MIC) 4 and Isavuconazole MIC 8), antifungal therapy with liposomal Amphotericin B was started. Also, a cutaneous septic embolization was observed on the legs and left arm, as confirmed by skin biopsy (Figure 1).

Hematological recovery (day 29) led to fever resolution and negative blood cultures. On day 35, BM evaluation showed morphological CR of the disease. On day 39, Tacrolimus was progressively reintroduced up to 1 mg/day on day 44, when the patient was discharged.

During post-induction follow-up, he started developing progressive pancytopenia. On routinary blood tests, 7000 copies of cytomegalovirus (CMV) were detected, and antiviral therapy with valganciclovir was started in the presence of increasing values. Meanwhile, the patient began to complain of unilateral visual impairment, initially interpreted as CMV retinitis, and he was readmitted to the Hematology Unit to intensify antiviral therapy with ganciclovir instead of valganciclovir. Due to further worsening of right eyesight, aqueous humor samples were taken, which resulted positive for fungi, but PCR analysis was unable to differentiate between *Aspergillus* and *Fusarium*. Systemic antifungal therapy was restarted with liposomal Amphotericin B and Isavuconazole, and it was optimized with intravitreal injections of Amphotericin B and voriconazole, because the presence of *Aspergillus* could not be ruled out. No other infectious foci were detected, and blood cultures yielded negative.

Shortly after, despite the need for intravitreal injection maintenance and systemic antifungal therapy and a dismal improvement of his sight, the patient, given the high risk of disease relapse, was admitted to the Transplant Unit to receive allogeneic HSCT of peripheral stem cells from an HLA-matched sibling donor.

As the hematopoietic cell transplantation comorbidity index (HCT-CI) was 6 (recent infection, ferritin levels > 2500, DLCOadjHb < 80%, albumin level < 3.5 g/dL), a non-myeloablative conditioning regimen with Treosulfan and Fludarabine was preferred over busulfan, given lower pulmonary toxicity and higher event-free survival and overall survival. The dosing schedule consisted of 10 g/m^2^ IV Treosulfan over days −4 to −2 and 30 mg/m^2^ IV Fludarabine (FLU) over days −6 to −2 [14].

Although BM stem cells were hypothesized as the best graft for their minor immunogenicity, unfortunately the identified HLA identical sister was inadequate for BM donation, and thus peripheral blood stem cells (PBSC) were selected as a graft, harvested through blood apheresis, after the administration of granulocyte colony stimulating factor (10 mg/kg for five days).

A total number of 332,06 × 10^8^ nucleated cells were infused: the total of CD34/kg was 5.2 × 10^6^, while the number of CD3 cells/kg was 1.07 × 10^8^.

The graft versus host disease (GVHD) prophylaxis consisted in IV Methotrexate 15 mg/m^2^ on day +1 and 10 mg/m^2^ on days +3 and +6, IV Tacrolimus to maintain a blood concentration of 10–15 ng/mL since day −6 and IV Thymoglobulin 2.5 mg/kg on day −1. The patient did not experience acute GVHD.

On admission, standard antiviral prophylaxis with acyclovir was started adjusting the drug dose based on the patient glomerular filtration rate; antifungal therapy with 200 mg daily Isavuconazole was continued. Weekly polymerase chain reaction (PCR) monitoring for CMV and Epstein–Barr virus (EBV) was performed until discharge from hospital and every other week during the first 100 days and at least monthly for another 9 months.

Since day −2, the patient complained of right eyesight worsening: he resumed Amphotericine B and voriconazole intravitreal injection and added daily 200 mg liposomial Amphotericin B to systemic antifungal therapy.

The early post-transplant period was characterized by a neutropenic fever of unknown origin (FUO), treated with broad-spectrum antibacterial therapy and a cutaneous rash responsive to topical glucocorticoid therapy, whose skin biopsy did not confirm an acute GVHD. No other organ toxicities occurred.

During the aplastic phase (day +8), the patient presented pain, edema, and conjunctival hemorrhage in the right eye. Orbital CT and MRI were performed, showing ethmoid and maxillary sinus inflammatory signs, and the patient upgraded systemic antibacterial therapy with IV Vancomycin and IV Meropenem. On day +27, a second facial MRI was performed showing resolution of inflammation but right retinal detachment.

The neutrophil (Ne > 500/mmc) and platelets (PLTs > 20,000/mmc) engraftment was reached on day +14 and +12, respectively (Figure 2).

Finally, the last aqueous humor sample performed on day +35 did not detect fungal infection (negative PCR), and Amphotericine B was suspended on day +40, whereas Isavuconazole was continued until day +82.

Prophylaxis for CMV reactivation with Letermovir was started at hospital discharge on day +47 and continued until day +150, and *Pneumocystis jirovecii* prophylaxis with Sulfamethoxazole/Trimethoprim was started on day +51 and continued at home.

A BM aspirate (day +40) showed a morphological CR and a blood donor chimerism (DC) of 97.6%, confirmed also at day +82 (DC 99.1%), day +190 (DC 98.3%), day +267 (DC 99.3%), and +365 (DC 99.2%).

About nine months after the transplant, pulmonologists recommended adding photopheresis to Tacrolimus after observing signs of early lung rejection, which the patient started with improvement of the pulmonary signs.

However, at 1 year and 4 months post-transplant, the patient started presenting leucopenia. Suspecting relapse, bone marrow evaluation was performed, showing 15–20% myeloid blasts with dysplastic features and marrow fibrosis. Molecular testing revealed a WT1 copy number of 2536, and chimerism shifted to 92% donor and 8% recipient.

Following confirmation of relapse, a multidisciplinary team decided to admit the patient to the Hematology Unit for salvage chemotherapy using the MEC protocol (Mitoxantrone, Cytarabine, and Etoposide). The hospitalization was complicated by a persistent fever of unknown origin. Post-reinduction bone marrow evaluation revealed persistent disease, and unfortunately, the patient died shortly after due to progressive disease.

## 3. Discussion

In managing this complex case, careful consideration was required in choosing the therapeutic course.

MDS/AML with TP53 is a high-risk AML type where HSCT is the main indication. However, HSCT is a complex, multistep, multifactorial, and highly individualized treatment associated with a significant risk of treatment-related mortality (TRM). As such, the indication for SCT required careful consideration, balancing the reduction in relapse risk (RR) against the potential for TRM [15,16].

The presence of a solid-organ transplant and the risk of possible rejection requires careful modulation of immunosuppressive therapy, contrasting the simultaneous need to reduce the risk of infection during post-chemotherapy aplasia. Moreover, the success of an HSCT depends on maintaining a graft-versus-leukemia response, which needs an active immune system, conflicting with lung rejection prophylaxis [3,17].

This already-intricate situation was further complicated by unusual infection sustained by a *Fusarium solani*, detected in blood cultures, with septic embolization and cutaneous involvement, and with ocular fungal infection, still under treatment at the time of the transplant. *Fusarium solani* is the most aggressive of *Fusarium* species and is able to give both superficial (keratitis and onychomycosis) and disseminated angioinvasive infections. Fungal infections are strictly related to immune competence, and this particular severe presentation was the consequence of a combined immunosuppression given by the chronic rejection prophylaxis and the acute leukemia with its treatment [18,19].

In this scenario, assessing TRM risk was particularly challenging due to a delicate balance of factors: the risk of rejecting the transplanted solid organ, necessitating chronic immunosuppression, the need to sustain a graft-versus-leukemia effect and thus an active immune system, and the risk posed by an active fungal infection, compounded by strict timing requirements to ensure the transplant’s efficacy.

First, we decided to manage the fungal infections with the multi-systemic approach utilized during the pre-transplant period using antifungal off-label treatments. Although posaconazole is the only licensed drug for antifungal prophylaxis during induction therapy for AML, because of patient’s infectious background, we used Isavuconazole. At the time of *Fusarium solani* infection, the antimycogram showed voriconazole resistance; thus, despite ambiguous indication for *Fusarium*, we preferred liposomial Amphotericin B to voriconazole, which is the only licensed drug for this infection treatment, at least in Europe. In the same way, we used Amphotericin B and voriconazole that are not licensed for intraocular administration. As clinicians, we overcame regulatory documents and real-life gaps, choosing these off-label strategies, to allow the patient to survive after increasing complexity treatment for AML.

Then, we administered a reduced-intensity conditioning with a personalized approach for the patient, choosing treosulfan [14]. After receiving HSCT, the patient did not experience TRM: the fungal infection was contained, GVHD was absent, and lung rejection was properly addressed. Despite permanent immunosuppression, progression-free survival lasted one year, and relapse was probably triggered by the need to intensify it with photopheresis to prevent lung rejection.

A complex case like this depicts how HSCT is a sophisticated and individualized approach that requires a deep understanding of both infectious diseases and immune responses, which is indispensable to optimize patient outcomes.

## 4. Conclusions

In conclusion, as survival rates for patients undergoing solid-organ transplantation (SOT) continue to improve due to advancements in surgical and clinical management, the complexity of cases that hematologists encounter is expected to increase. Post-transplant complications have become a significant challenge, particularly with the elevated risk of cancer development linked to prolonged immunosuppressive therapy. This risk can reach up to 40% within 20 years following transplantation. Managing infections in patients who have received a solid-organ transplant and are undergoing hematopoietic stem cell transplantation requires a delicate balance between protecting against pathogens and carefully modulating the immune response to prevent relapse of hematological disease. Effectively managing these infectious complications is essential to optimize patient outcomes and reduce long-term mortality [2,17].

## Figures and Tables

**Figure 1 microorganisms-13-00703-f001:**
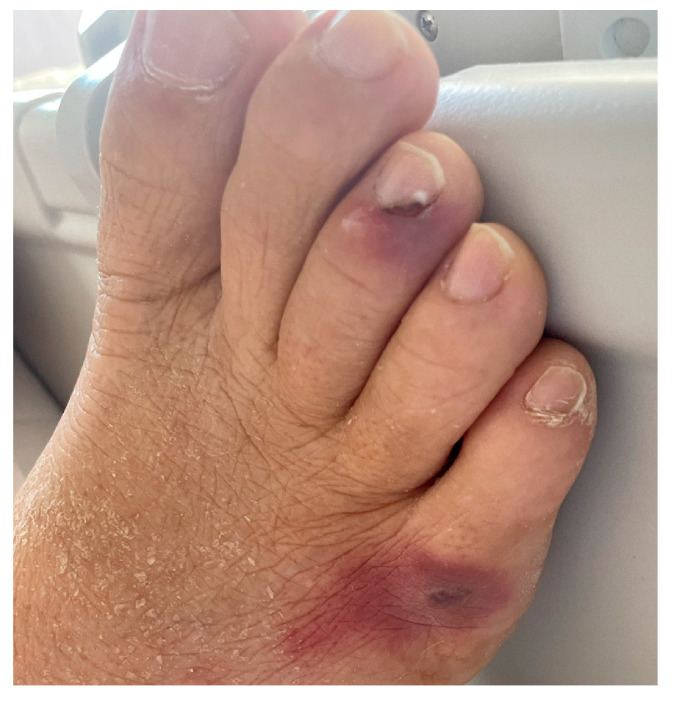
*Fusarium solani* cutaneous septic embolization.

**Figure 2 microorganisms-13-00703-f002:**
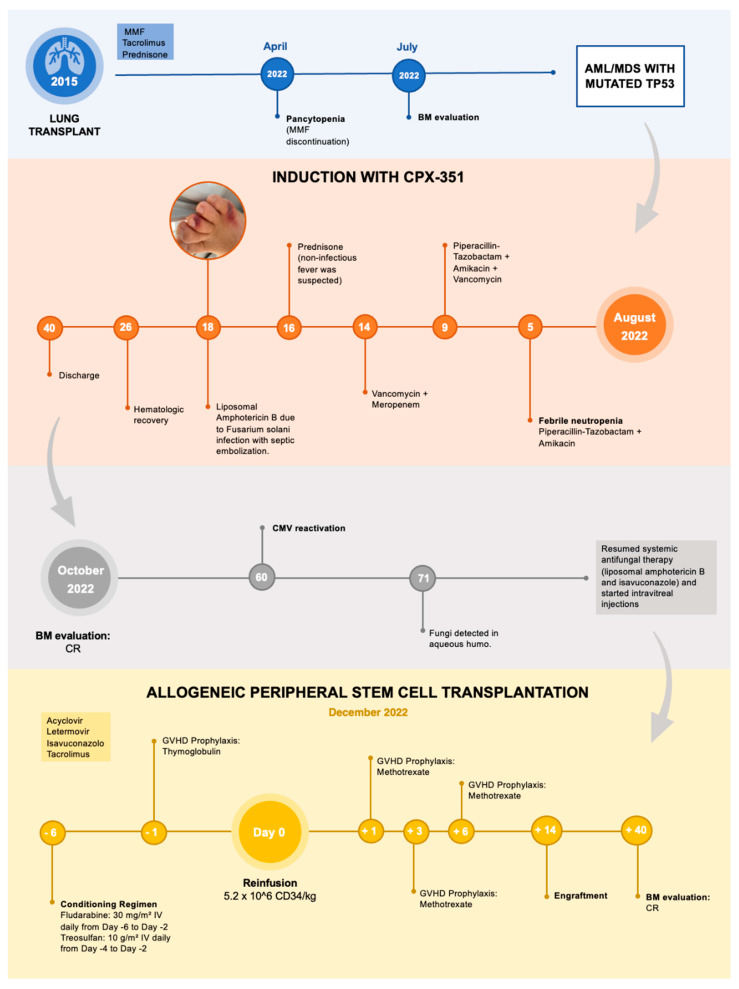
Short summary of patient medical history.

## Data Availability

All relevant data are contained within the article.

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
