# Peer review of "Allogeneic Hematopoietic Stem Cell Transplantation Despite Severe Fusarium solani Infection in a Lung Transplanted Patient—A Case Report"

_microorganisms, 2025, doi:10.3390/microorganisms13040703_

Round 1

Reviewer 1 Report

Comments and Suggestions for Authors

The authors present an interesting clinical case, where a patient, previously undergoing lung transplantation, developed secondary high-risk AML, complicated by Fusarium infection during the induction treatment. In spite of the very high-risk clinical profile, the patient successfully underwent allogeneic HSCT from a sibling donor. Unfortunately, he eventually died of AML relapse. GENERAL REMARKS - The reported case is very interesting because of the efforts to manage an extremely complex and contradictory clinical case. Paradoxically, the most criticizable section is the title, “Severe fungal infection in a lung transplanted patient who received allogeneic hematopoietic stem cell transplantation: first reported case in the literature.”. The reader could assume that fungal infection developed after HSCT; whereas the manuscript clearly reports something else. Moreover, the authors should specify also in the title, that Fusarium solani was the causative agents of the “severe fungal infection”. -The authors focus on reporting a “first case”, whereas the key point is their ability of successfully performing allogeneic HSCT after a recent invasive fungal infection, in an otherwise very high-risk patient. - In the discussion, the authors should pay more attention to comment their therapeutic choices, eventually allowing the patient to survive after HSCT. The interest for the matter is accrued by the intolerance to posaconazole and by the resistance of the causative agent to voriconazole. - Since posaconazole is the only licensed drug for antifungal prophylaxis during induction therapy for AML, since voriconazole is the only licensed drug for the treatment of Fusarium infection, at least in Europe, the reported treatments were administered off-label. The indications for liposomal amphotericin-B are ambiguous and do not include Fusarium; the same goes for isavuconazole. Moreover, isavuconazole is not licensed for prophylaxis. Finally, neither amphotericin-B nor voriconazole are licensed for intraocular administration. In my opinion, the authors could comment the gap between the regulatory documents and real life, where clinicians are called to allow patients to survive after treatment strategies of increasing complexity. MINOR POINTS 57 Lung transplant was complicated by severe infections. The authors could offer some more detail. 60. WBC are interesting, neutrophils even more. 65. Bone marrow biopsy? 81. Progressively suspended? 78. “Well tolerated” is an annoying locution, frequently retrieved from clinical records. In this case, it is also questionable, since the patient developed fever on day 5. 79. Intolerance to posaconazole should be presented in more detail. Moreover, its possible role in favoring a deep fungal infection, could be discussed. 85. I do not understand the choice of Piperacillin/tazobactam in tailoring antibiotic therapy to ESBL+ E. coli. 94. Resistance to azoles (voriconazole, likely not isavuconazole) should be presented in more detail. Moreover, its possible role in favoring Fusarium ocular localization, could be discussed. 105. Starting valganciclovir with 7000 copies is legitimate, notably in the presence of increasing values and retinal impairment. Nevertheless, the authors should better clarify. 110. Fungi, no more. 112. Why intravitreal voriconazole in spite of resistance? 124. The sentence sounds rather strange 125 Inadequate. Why? The most obvious explication could be weight mismatch; but the patient was a low weight man, as far as it can be retrieved from CD34+ infused cells. 126. Mobilization schedule. 127. Why cryopreserved? 149. Massive facial MRI? - Viral prophylaxis is presented on line 134 and continued on line 156, with no relevant events in the meanwhile. The matter could be condensed in a single paragraph. 173. Passed away shortly after (Figure 2), rather colloquial. Moreover, Figure 2 does not present relapse and death. It should be recalled elsewhere.

Reviewer 2 Report

Comments and Suggestions for Authors

The manuscript titled “Severe fungal infection in a lung transplanted patient who received allogeneic hematopoietic stem cell transplantation: first reported case in the literature” presents a novel and clinically significant case. It describes the management of a 55-year-old male patient who underwent allogeneic hematopoietic stem cell transplantation (HSCT) after a bilateral lung transplant. The case is particularly valuable as it highlights the challenges of balancing immunosuppression, infection control, and disease management in a highly complex medical scenario.

I also have the following comments:

- i suggest an extension of the introduction.

- please elaborate the choice of treatment, how and why were those drugs chosen. Is there an established protocol, if yes, is it specific for the hospital/ country or is it international?

- regarding English, I suggest some minor revisions, especially in the introduction. (example. "Over time, hematologists have developed a comprehensive understanding of common infectious agents, leading to implementation of well-defined protocols for early and effective management of febrile neutropenia, reducing infectious related mortality (IRM)." uses lenghty and redundant phrasing)

Comments on the Quality of English Language

Regarding English, I suggest some minor revisions, especially in the introduction. (example. "Over time, hematologists have developed a comprehensive understanding of common infectious agents, leading to implementation of well-defined protocols for early and effective management of febrile neutropenia, reducing infectious related mortality (IRM)." uses lenghty and redundant phrasing)

Reviewer 3 Report

Comments and Suggestions for Authors

The article entitled "Severe fungal infection in a lung transplanted patient who received allogeneic hematopoietic stem cell transplantation: first reported case in the literature" is a case report on a rare systemic infection with Fusarium solani. I do believe that the article is original and presents a very rare case of infection which would benefit clinicians greatly if published.

However, I would like to suggest some improvements:

- can you please remove the "." at the end of the title?

-the Introduction section, in my oppinion, needs to be revised. The text is too general and it does not bring enough information to the reader about the main topic of the article. Furthermore, there are paragraphs that do not have a reference mentioned (e.g. first paragraph).

What I would suggest adding to the Introduction section is a paragraph summarizing some general things about Fusarium spp., the entry pathway of this fungi and the most common diseases it causes, some species that are common in pathology, risk factors for acquiring the disease, the incidence (in your country, in general, in hematologic patients, etc.). You can also add some general information about AML/MDS with TP53 mutation, its incidence in the area, how common fungal infections are in those patients, etc. After all, the main objective of this case presentation is to bring into light a very rare case of infection, and I think some details about incidence would give the reader an idea on why they should keep reading the article

- line 93 - please write Fusarium solani in italic

- please check all microorganism names in the text and write them in italic (e.g. line 157 - P. jirovecii, line 58 - S. pneumoniae and P. aeruginosa, etc.). I would also suggest, if possible, to write the full name of each microorganism, especially since most of them are only once mentioned in the text

- can you please rotate Figure 2 in order to make it more easily readable? Alternatively, you can also put the page in Landscape format to make the figure fit, then rotate it with the text in the right direction.

- line 187 - "post-cht" - did you mean post-chemoterapy? Please write the full word

- what do you think was the starting point of the systemic infection with Fusarium solani? I saw some disseminations mentioned throught the text (skin, eye), but it was never clear which was the portal of entry for the fungi or if the patient had any symptoms related to other organs before developing the systemic infections?

- the discussion section needs extensive revising as well, in my oppinion. At the moment, it does briefly describe the case and the treatment applied, but I think it should also focus on comparing the current case with the existing literature (e.g. are there any more cases like this described in other articles? What was the treatment applied there and how was it different from the one applied in the current case? What makes the patient in this case special from others described in literature?)

- Regarding the Conclusion section, I think this should also undergo revising. As I see it, the article focuses on a rare case of diseminated fungal infection with a rare pathogen (Fusarium solani) in a patient who underwent HSCT due to AML/MDS. That is what the article should focus on. It is indeed a very complicated case with a very complex patient that required multiple board meetings with different specialists for a clear treatment, but at the same time, there are a lot of different cases just as complex. Mentioning in the conclusion that hematologists will just have to treat more and more complex cases is something that is generally known (or I think should be expected in the current days with all the technological progress). The focus on the conclusion section should be to briefly summarize the case, the treatment followed, what worked or not, what this case report brings as a novelty, keeping the general statements to a minimum

- nothing is mentioned in the article about Ethical Committee approval, and since this is a case report, I think it is mandatory. Can you please revise?

Comments on the Quality of English Language

Please revise the English language. There are several places throught the text where words are incomplete or the verb is in the wrong tense.
